# Toward an Adaptive Threshold on Cooperative Bandwidth Management Based on Hierarchical Reinforcement Learning

**DOI:** 10.3390/s21217053

**Published:** 2021-10-25

**Authors:** Motahareh Mobasheri, Yangwoo Kim, Woongsup Kim

**Affiliations:** Information and Communication Engineering Department, Dongguk University, Seoul 04620, Korea; mm.mobasheri@dongguk.edu (M.M.); woongsup@dongguk.edu (W.K.)

**Keywords:** internet of things, fog computing, fog fragment cooperation, hierarchical reinforcement learning

## Abstract

With the increase in Internet of Things (IoT) devices and network communications, but with less bandwidth growth, the resulting constraints must be overcome. Due to the network complexity and uncertainty of emergency distribution parameters in smart environments, using predetermined rules seems illogical. Reinforcement learning (RL), as a powerful machine learning approach, can handle such smart environments without a trainer or supervisor. Recently, we worked on bandwidth management in a smart environment with several fog fragments using limited shared bandwidth, where IoT devices may experience uncertain emergencies in terms of the time and sequence needed for more bandwidth for further higher-level communication. We introduced fog fragment cooperation using an RL approach under a predefined fixed threshold constraint. In this study, we promote this approach by removing the fixed level of restriction of the threshold through hierarchical reinforcement learning (HRL) and completing the cooperation qualification. At the first learning hierarchy level of the proposed approach, the best threshold level is learned over time, and the final results are used by the second learning hierarchy level, where the fog node learns the best device for helping an emergency device by temporarily lending the bandwidth. Although equipping the method to the adaptive threshold and restricting fog fragment cooperation make the learning procedure more difficult, the HRL approach increases the method’s efficiency in terms of time and performance.

## 1. Introduction

With the advent of cloud and fog computing, as a cloud complement, and the following emergence of the Internet of Things (IoT), data generation has become faster through various smart environments. Over time, with the increasing speed of data generation, the term *big data* has emerged in network concepts. In contrast, network bandwidth improvement has been insufficient to cover the increase in IoT devices and the volume of generated data; therefore, the bandwidth limitation is a crucial IoT challenge and should be considered in studies as a serious constraint.

Despite the advent of the IoT with its useful capabilities for network applications and equipment augmentation due to user demand growth, some limitations must be considered [1]. One is the bandwidth limitations, which cause bottlenecks in network communication [2]. Accordingly, it is essential to pay attention to the network bandwidth in smart environments, such as smart homes [3,4,5], smart cities [6,7], smart factories [8,9,10], healthcare [11,12,13], smart metering [14], robotics [15,16], energy management systems [17], and industrial IoT (IIoT) [18,19,20], because numerous kinds of devices cooperate using heterogeneous network communication [21,22]. The steady increase in network complexity and the sharing of physical resources, such as the network bandwidth, leads to flexible and efficient resource management approaches [23].

In this way, the authors in [24] presented a dynamic approach to bandwidth management in networked control systems in which control loops use the bandwidth according to the dynamics of the controlled process while attempting to optimize the overall control performance. The authors in [25] focused on managing real-time flows and predicting the amount of bandwidth needed by a mobile host during its movements using a Markovian approach. The resource allocation procedures and weighting policy were designed using a software-defined network (SDN) controller to perform bandwidth management [26]. This mechanism monitors and analyzes networks and the application’s status. Then, based on the results, it reallocates resources dynamically with a software-defined discipline.

Adaptive bandwidth management was designed via a prototyping technique to increase the bandwidth of institutional, educational users who access more on educational websites [27]. The problem does not include fog node cooperation to increase efficiency, and the proposed technique should recognize target users who are worthy of receiving more bandwidth. The authors in [28] improved fault resilience and managed the required bandwidth in industrial cyber-physical system networks through SDN. The problems of shared bandwidth and the size of the network expansion were studied in [23]. The authors focused on the demand for flexible and efficient communication network management in IIoT applications while different flow types share the bandwidth. They proposed a resource management mechanism using a combined SDN and network virtualization solution under varying workloads and flow priorities to reduce network management complexity. This approach exploits a priority-based runtime bandwidth distribution mechanism to dynamically react to load changes due to alarms. The smart environment is not fragmented, and the proposed mechanism seems usable for one smart fragment only. Therefore, it is not useful for a scenario where several cooperating smart fragments exist, each with its fog node. Moreover, the authors in [23] used the SDN and network visualization, where the control plane does not need to be disassociated from the data plane, if machine learning techniques were used.

The vast smart environment of [29] includes several environments called fragments. Each fragment has several IoT devices and a single fog node as the fragment manager responsible for communicating with a higher-level manager (e.g., a cloud server). In an emergency detection by an IoT device, it informs its connected fog node. The amount of assigned bandwidth for each IoT device is fixed; thus, the fog node is responsible for assigning the bandwidth of another IoT device as the emergency device requires and for taking it back when the emergency is over. Under the conditions with a fixed predefined threshold, the fog node can borrow the bandwidth of a neighboring fragment’s feasible IoT device. The fog cooperation problem is solved via the RL approach, where the received reward of the learning procedure is based on a fixed predefined threshold.

Due to the development of learning cases and the surge in state and action spaces, the typical RL is not an appropriate approach. We use hierarchical RL (HRL) as a complement to the typical RL approach. The process of learning the threshold is performed at the first hierarchy level, and the second hierarchy learning level starts based on the results of the first learning level. The fog node starts to learn to select the best helper for an emergency device among its own and neighbors’ feasible help lists based on the learned threshold, its situation, and neighboring situations.

In this study, at the first level, the fog node should learn the best level for the threshold based on internal and neighboring situations, and at the second level, it should learn to select the best helper among internal and neighboring nodes by following the rules. The main issue is fog cooperation with a high-performance guarantee in a limited bandwidth network; thus, the neighbor fragment situations must be considered (the number of neighbor emergency devices) in cases where the fog node wants to use their bandwidth. The predetermined fixed threshold is a definite constraint in fog cooperation; therefore, it causes trouble in the cases allowing a fog node to receive help from a neighbor while that neighbor’s situation is not normal. In general, helping others should be achieved when no internal emergency problems and subsequent bandwidth requirements exist.

In the remainder of this paper, the research problem and solution approach are explained in detail in Section 2. In Section 3, we formalize the decision-making problem and describe the simulation results of the proposed approach, followed by the conclusions in Section 4.

## 2. Materials and Methods

In RL, the agent’s goal is represented through a reward function representing a special signal received from the environment by the agent, and its value varies for each step. Moreover, the agent should maximize the total amount of the received reward (i.e., maximizing the immediate reward and long-term cumulative reward). Formalizing an agent’s goal through a reward signal is one of the most distinctive features of RL [30]. An RL agent maximizes its cumulative reward through interaction with an unknown environment (i.e., the agent observes a state, takes action, receives an immediate reward, and transmits to the next state, repeating this cycle in each step). The agent is supposed to learn the behavior, called the policy, to map the observed states to the appropriate actions. To maximize the expected future sum of rewards, the agent must explore the environment by taking various actions and observing their effects, and after a while, exploiting its current knowledge through selecting actions with the highest past rewards. An essential challenge in RL is solving domains with sparse rewards (i.e., when most of the immediate reward signal is zero). In this case, all actions initially appear equally good, and it becomes crucial to explore actions efficiently until the agent finds a high-reward action. Therefore, it becomes possible to distinguish actions that eventually lead to the highest rewards [31].

Although RL has been widespread and achieved superhuman performance in various fields, such as system management and configuration, it has some problems [32,33,34,35], such as scaling (i.e., when the state and action spaces increase, the efficiency decreases). Several approaches have been developed to manage RL problems, such as HRL, a computational approach intended to address scaling by learning to operate at various temporal abstraction levels. Unlike RL with just one policy to achieve the goal, the HRL approach has several subpolicies that work together in a hierarchical structure. In this technique, the policy is divided into several subpolicies, and the actions of the higher-level policies are passed to their lower levels as their goals over smaller time horizons [32,35]. Therefore, one of the benefits of HRL is improved exploration because the action space is reduced, and exploration is simpler.

Moreover, HRL learns a policy comprising multiple layers responsible for control at each temporal abstraction level. The key novelty of HRL is to extend the available action set so that the agent can choose to perform primitive and macro actions (i.e., sequences of lower-level actions [36]). In addition, HRL facilitates exploration because the number of necessary decisions is reduced before obtaining a reward [31]. In the control aspect, a macro is an open-loop control policy and is inappropriate for stochastic system control. Further, HRL approaches generalize the macro idea to closed-loop policies or closed-loop partial policies, as they are defined for a subset of the state space. Furthermore, partial policies must have well-defined termination conditions [37].

The concurrency of multiagent coordination is the basis for modeling coordination among multiple simultaneous agents. From a theoretical aspect, it matters whether concurrent actions are executed by a single agent or multiple cooperating agents. One of the complexities of the multiagent problem is that an agent cannot generally perceive the actions or states of other agents in the environment (hidden states). The authors in [37] introduced a learning policy across joint states and actions and assumed that states indicate the joint state of *n* agents, where each agent *i* may only have access to a partial view *s_i_* and may not know the other agents’ actions. A separate connection between HRL and multiagent RL has been studied in [35]. In this study, multiple fog nodes of the scenario are set up as a multiagent system. As mentioned in the previous section, the RL approach of [29] employs a fixed-value threshold. In this study, by adding the ability to vary the threshold level (variable threshold), we break the policy into two subpolicies and proceed using HRL.

The authors in [38] introduced a decision-making problem of a single fog node connected to various IoT devices, determined via different priorities. The fog node belongs to the only smart fragment of a small environment. This scenario is extended to a vast smart environment with various fog nodes, and each belongs to its distinct fragment [29]. Then, these fog nodes are to cooperate so that the fragment and cooperation conditions are met.

When a device encounters an abnormal situation, it is responsible for informing its connected fog node of the situation, and the fog node is responsible for making it possible to receive more data from this emergency device, whereas the assigned transmission bandwidth is insufficient. In an emergency, its assigned bandwidth may not be sufficient for an emergency device due to the limited network bandwidth, and some devices in a normal situation should decrease the bandwidth to provide additional bandwidth capacity for emergency devices. Moreover, as the number of emergency devices increases, extra bandwidth assignment becomes more complicated.

This issue considerably affects the environmental network when saving a millisecond is essential for preventing damage and failure. The fog node is supposed to learn the best selections among devices in ordinary situations to solve this problem and help devices in emergencies by decreasing their transmission rates to the fog node. In the proposed approach of [29,38], the fog node acts as an agent of the RL approach. Although the fog node does not have a trainer or information regarding the distribution parameters of future events, it can successfully gain an optimal policy through RL. As the learning process converges, the fog node decisions become future-oriented and optimal. The learned selections of the fog node help the system face the minimum number of devices with a lack of bandwidth.

Large smart environments are fragmented, with many smaller smart environments with their designated fog nodes. As the total bandwidth is limited and shared among these fragments, the fog node cooperation is raised in [29], and each smart environment is assumed to be a fog fragment; that is, each fog node and its connected IoT devices are called a fog fragment. Therefore, in addition to learning the best helper for an emergency device using the related fog node (as discussed in [38]), the cooperation of these fragments in using a shared limited bandwidth under special conditions should be learned by fragments’ fog nodes, as discussed in [29].

The learning procedure for each fog node is the same as in [38]; therefore, the fog node learns the best helper among devices with a normal situation and sufficient bandwidth (called feasible devices) based on the device priority. The reason is that selecting a higher-priority helper is not logical and efficient when a device with a lower priority exists for helping. Moreover, fog nodes are allowed to receive help from their neighboring fog nodes under defined conditions. A predefined threshold exists for each fragment that is set in the priority range of its devices. The fog node can receive help from a neighboring fragment by borrowing the bandwidth of a device in that neighbor’s help list only if no feasible device is available with a priority lower than the fragment threshold. The threshold levels of the fragments are predefined and fixed in [29]. The fragment situations change over time; thus, the threshold must be dynamic.

In this study, the fragment thresholds are learned to be used in the subsequent learning of the fog nodes (i.e., managing the bandwidth among normal devices and emergencies through HRL). In addition, HRL decomposes an RL problem into a hierarchy of subproblems or subtasks so that the higher-level problem invokes lower-level problem tasks as if they were primitive actions [32]. Consequently, each fog node should learn three objects: first, the best helpers for handling the required extra bandwidth for connected emergency devices; second, the conditions of receiving help from neighboring nodes; and third, learning the best threshold level based on external (neighbors) and internal situations.

The fog node should update a variable, *z_m_*, in every time step to determine the fragment situation and inform its neighbors about it. Moreover, each fog node should calculate another variable, *z_n_*, as the average of *z_m_* of its neighbors. Based on *z_m_*, *z_n_*, and the last time step threshold (*Th_old_*), the fog node learns the best threshold in the current time step through RL (Figure 1A). The threshold level can be selected from the range of green to orange (details in Section 3.1). In each time step, a matrix including the last selected threshold, *Th_new_*, based on *z_n_*, *z_m_*, and *Th_old_*, is updated (a further explanation is provided in Section 3.1.2). When the threshold learning procedure is completed in the first fog node learning level, the next learning level starts based on the results of the first learning level. The fog node starts to learn by selecting the best helper for an emergency device among its own and neighbors’ feasible help lists among its feasible connected devices and the neighbors’ feasible lowest priority devices based on the index of the emergency device (*i*), *z_n_*, *z_m_*, and *Th_new_*, extracted from the final *threshold* matrix (Figure 1B). Therefore, the fragment bandwidth varies by emergency in different fragments. As the result of two-level learning, fragment cooperation leads to dynamic bandwidth allocations for each fragment in every time step, whereas the sum of bandwidth allocations for all devices is fixed (Figure 1C).

As discussed in [29], the threshold value is specified at the beginning of smart city life through a smart city management configuration to specify the boundaries of obtaining help from neighbors mentioned in the help lists. Therefore, the fog node is encouraged to select a helper from its fragment when a feasible device exists with a lower priority than the threshold. Otherwise, the fog node must select a helper from other fragments in its vicinity. If an emergency occurs in a fragment without a feasible device with a priority lower than the threshold, the fog node can use the neighbors’ help lists and select a helper with adequate bandwidth for the emergency device. All these rules are learned over time through RL.

Figure 2 illustrates a simple example of forming a feasible device set by the learner. The first part of this figure, A, represents all IoT devices in the learner fragment, specified with white, green, yellow, orange, and red as their priorities. Assuming the learner is in the early stages of learning and that all devices have sufficient bandwidth, all except the red priority devices (highest priority devices) can be helpers. Part B determines these possible helpers by dashed-line squares. If the threshold level is yellow (*Th_new_* = 2) and one of the orange priority devices has an emergency, only devices with white and green priorities should be included in the feasible device set. These devices are distinguishable in Part C.

In this study, having no feasible device with a priority lower than the threshold is not the only condition for obtaining help from a neighbor. It is necessary to consider the neighbors’ and internal situations. When a neighbor experiences a bad situation, it is not logical to borrow the bandwidth of one of its help-list devices because that neighbor should primarily handle its internal situation.

The authors in [29] appropriately defined a fixed threshold value to ensure a proper tradeoff between choosing from its own or neighboring feasible devices. Defining a fixed threshold value (or fixed rules for it) is not a proper approach. It is better to encourage or punish the fog nodes as an RL agent to learn the optimal policy to increase, decrease, or maintain the current threshold level. Therefore, the fog node should learn the threshold level and best device for helping an emergency device over time without instructions from a supervisor. Based on the RL approach, each fog node attempts all feasible threshold levels one by one in each visit of all combinations of different situations that a fragment and its neighbors may experience without considering any condition. In other words, the fog node is free to test all threshold levels in different situations of itself and the neighbors. After each selection, the fog node receives a reward value, indicating the quality of its recently selected threshold. Therefore, the fog node learns the best threshold level over the learning process through the received rewards and applies the best experience for successive similar visits in the future without a trainer or supervisor.

As fully discussed in [29], the threshold value is specified at the beginning of the smart city life through smart city management configuration in order to specify the boundary of obtaining help from neighboring helpers, mentioned in their sent help lists. Therefore, the fog node is encouraged to select a helper from its own fragment when there is a feasible device with a lower priority than the threshold in its fragment. Otherwise, the fog node has to select a helper from other fragments in its vicinity. In other words, if an emergency takes place in a fragment without any feasible device with a priority less than the threshold, the fog node is allowed to use the neighbors’ help list and select a helper with adequate bandwidth for the recent emergency device. Note that all these rules are learned over time through RL.

In this study, having no feasible device with priority less than the threshold was not the only condition of receiving help from a neighbor. It is necessary to consider the neighbors’ situations in addition to the internal situation. In fact, when a neighbor is experiencing a bad situation, it is not logical to borrow the bandwidth of one of its help list devices, since that neighbor should handle its internal situation primarily.

Assumptions of this study are as follows:Each fragment has only a single fog node, and all IoT devices in a fragment are connected to the fog node of that fragment;In each time step, all fog nodes send their help lists, including their situation indicators (*z_m_*) and the lowest priority devices with their current bandwidth to all neighboring fog nodes;Fog node cooperation is performed through a dedicated wireless interface; therefore, fog node communication performs at disparate frequencies from those assigned for fog node communication with connected IoT devices;As with [29,38], eliminating emergencies is not studied in this paper, and the focus is on managing the bandwidth in emergencies as higher-level decisions require more bandwidth to receive more information.

## 3. Modeling and Results

As mentioned, several fog fragments exist in a vast smart environment, and they operate in the same way. Therefore, we focus on one fog node in its fragment in the vicinity of other fog nodes and distinguish it from the neighboring fog nodes by calling it the learner fog node or “the learner.” The learner’s decision-making problem is modeled in [29] concerning selecting the best device among its own feasible devices and the neighbors’ feasible lowest priority devices to help an emergency device so that the reward and punishment values are maximized and minimized, respectively. In this study, the model features are completed in the second level of the learner’s learning, and before this level, the threshold is made adaptive and variable as a learning procedure that should be performed before starting the second learning level. Therefore, the hierarchy in these learning procedures includes learning the best threshold considering the internal and external situation as the first or lower learning hierarchy level and learning the best helper based on the learned threshold as the second or higher learning hierarchy level. We explain the first, then the second level below.

### 3.1. First Learning Hierarchy Level: Learning the Best Threshold Value

Each fragment threshold level is in the range of its IoT device priorities. The threshold level should be between the lowest and highest priorities. If the threshold is adjusted to the highest priority, the fog node cannot obtain help from neighbors, and if it becomes equal to the lowest priority, the fog node selects a helper among neighbors’ lowest priority devices, even in cases with feasible devices with higher priorities than the fragment threshold. Therefore, the threshold should be appropriately adjusted to ensure a proper tradeoff between choosing from its own or neighboring feasible devices. Unlike [29], the threshold value is not fixed and is realized by learning by receiving rewards. A variable threshold or adaptive threshold indicates that the threshold constraint of the learner’s threshold is changed based on internal and neighboring situations.

In contrast to [29], we consider colors instead of numbers to illustrate the priority levels. White represents the minimum priority, and green represents a low priority higher than the white priority. Yellow represents a medium priority higher than the green priority, and orange represents a high priority higher than the yellow priority and less than the red priority, which is the highest. The threshold level can be switched among the range of green to orange and cannot be equal to the white priority because the fog node would always be allowed to obtain help from neighbors, even when some devices in its feasible device set have low priorities. Additionally, it cannot be equal to red priority because the fog node cannot receive help in any situation. Therefore, the threshold level could be green, yellow, or orange, and level changing is performed by learning in the first level of HRL. For this purpose, the state, action, policy, and reward functions are defined as follows.

#### 3.1.1. State

The states of learning the best threshold level should describe the exact situation of the fragment, as it is the basis of learning. Therefore, the state should consist of neighbors and internal situations, in addition to the last threshold. The combination of *z_n_*, *z_m_*, and *Th_old_* (last threshold) satisfies this need. Although the goal of [29] was to handle the needed bandwidth in emergencies to receive more data from emergency devices, the focus was on the number of emergency devices: the number of normal devices plus the emergency devices that received help was calculated in the reward function as a model for determining the situation of the fog node fragment. In this study, an indicator representing the fog fragment situation (the inner situation) is needed. Moreover, *z_m_* is a weighted average representing the internal situation and is calculated in Equation (1) as follows:(1)$$z_{m}=Ω\left(\;\frac{\sum_{i=1}^{n_{d}}flag\left(i\right)\times p\left(i\right)}{\sum_{i=1}^{n_{d}}(p\left(i\right)|\;flag\left(i\right)=1)\;}\right),$$
where *p(i)* denotes the priority of device *I,* and *flag(i)* determines the situation of device *i*, where *flag* is a Boolean array presenting the situations of *n_d_* connected devices to the learner as 1 and 0 for emergency and other devices, respectively, based on their indices. Moreover, Ω is a function that maps$\;z_{m}$ to either 0 or 1 using a constant *c* (e.g., when *c* = 0.5, values below 0.5 are approximated to 0, meaning the internal situation is good or normal, whereas values above 0.5 are rounded to 1, meaning the internal situation is bad or in an emergency).

The external situation (neighbor situation) is another influential factor that should be considered when borrowing bandwidth from a device out of the fog node fragment. This indicator is denoted by $z_{n}\;$and is equal to the weighted average of the learner’s neighbors’$z_{m}$, where the number of neighbors is denoted by *n_b_* in Equation (2) as follows:(2)$$z_{n}=Ω\left(\frac{\sum_{j=1}^{n_{b}}z_{m}\left(j\right)}{n_{b}}\right),$$
where *z_n_* should be mapped to either 0 or 1, the same as *z_m_*. Therefore, if *c* = 0.5, values below 0.5 are approximated to 0, meaning the external situation is good or normal, whereas values above 0.5 are rounded to 1, meaning the internal situation is bad or in an emergency. Moreover, *z_n_* and *z_m_* respectively indicate the internal and external situations affected by accidental events, and the fog node does not have any information about the distribution parameters of future emergencies.

#### 3.1.2. Action

Based on *z_n_*, *z_m_*, and the last threshold (*Th_old_*), the learner should decide on the new threshold level (*Th_new_*). Therefore, the action represents the determined color level for the threshold. A matrix called *threshold* is updated by the fog node based on its visited state and selected threshold through RL in every time step, in Equation (3):(3)$$Threshold\left(z_{n},z_{m},Th_{old}\right)=Th_{new},$$
where the selected level for the next threshold by the fog node is inserted as a member of the *threshold* matrix with the position of *z_n_*, *z_m_*, and *Th_old_* as the first, second, and third dimensions, respectively. Transference among different states is not only through the selected actions; $z=[z_{n}\;z_{m}]$ is the primary indicator for deciding on the action that will become one of the state’s items for the next time step.

#### 3.1.3. Policy

A learning policy defines the learner’s behavior at a given time (i.e., a policy determines how to map the perceived states of the environment to the proper actions). One of the policy rules in this study is that the increase or decrease in the threshold level is step by step (e.g., when the threshold level is green and should be increased, it becomes yellow, and if more increase is needed, it becomes orange in the next time step). Table 1 describes the learning policy in detail, in which 0 and 1 in the policy columns indicate yes and no, respectively (e.g., according to Equations (1) and (2), when the internal and neighbors’ situations are equal to 0, they are fine (row 1), thus decreasing the threshold level and helping others are allowed). If the internal and neighbor’s situations are both bad (row 4), the threshold is not decreased, and helping others is not allowed or logical because lowering the threshold leads to more reliance on the neighbors’ helpers during their emergencies.

Based on Table 1, Figure 3 is obtained, illustrating the transition among threshold levels based on the perceived situations of the fragment and neighbors. The threshold levels of green, yellow, and orange are numbered 1, 2, and 3, respectively, and $z_{n}$ and $z_{m}\;$are represented by the array$\;z=[z_{n}\;z_{m}]$. The primary threshold level for all fog nodes is yellow, and when the situations for all fragments are fine, the fog nodes should return to this primary level step by step.

#### 3.1.4. Reward Function

As mentioned, defining the reward function is the most important part of an RL approach because it presents the effects of the learner’s actions on the environment and the fog node fragment. Moreover, the reward function should be defined to satisfy the learning policy goal. Based on the learning policy, Table 2 lists the policy goals in detail. For example, when the internal and neighboring situations are fine (row 1), the target level is yellow (level 2) regardless of the last threshold level. When the internal and neighboring situations are both bad (row 4), if the last threshold level was 1, the target level is 1, and if the last threshold level was 2 or 3, the target level is 3.

We define the reward function based on the policy goal. As explained, an RL agent takes action randomly, observes the received rewards of actions, and learns the best action in each state using this procedure. As the learning is reinforced over time, the probability of random action selection decreases. According to Table 2, when the learner selects a correct action, it receives a reward value, and when it takes an incorrect action (the wrong level for the threshold), the received reward value is 0. Equation (4) presents the reward function of the first learning level *R_Th_* (i.e., the reward for learning the best threshold level):(4)$$R_{Th}\left(n\right)=N_{Th}-the\;number\;of\;nonzero\;elements\;of\;\left(Threshold-Th_{c}\right),$$
where *Threshold* is a matrix of the learner’s experiences and *Th_c_* is the target threshold matrix from Table 2. In addition, *N_Th_* is calculated via Equation (5) and provides the number of threshold elements:(5)$$N_{Th}=2^{Th_{l}}\times \left(2^{Th_{l}}-1\right),$$
where we equate the threshold length (*Th_l_*) to 2 because two bits are needed to represent three levels $\left(2^{Th_{l}}-1\right)\;$for the threshold (green, yellow, and orange). The algorithm of the first fog node learning level is in Algorithm 1.

**Algorithm 1:** Learning the best threshold level in the first learning hierarchy level.**1.** Input:  Initialize time step (*n*), α, γ,$Th_{l},\;Th_{old}$, *Threshold*, and $Q_{Th}$ matrices.  Make $Th_{c}\;matrix$ as the accurate threshold matrix based on the transition states.**2.** While not converged, do **1.** Generate two binary random values as $z_{n}\;and\;z_{m}$ **2.** Determine the accurate threshold, *Th*, based on $Th_{c}$ **3.** With probability ɛ, randomly choose a value for the threshold from $\left(2^{Th_{l}}-1\right)$ levels. **4.** Otherwise, $Th_{new}=argmax_{j\in \;\left(2^{Th_{l}}-1\right)\;}\left(Q_{th}\left(Th_{old},j\right)\right)$ **5.** If $Th=Th_{new}$   $R_{Th}\left(n\right)=$   $N_{Th}-the\;number\;of\;nonzero\;elements\;of\;(Threshold-Th_{c}$) **6.** else    $R_{Th}\left(n\right)=0$ **7.**  End **8.** Update the $Q_{Th}$ matrix    $Q_{Th}\left(z_{n},z_{m},Th_{old},Th_{new}\right)=\left(1-\alpha \right)Q_{Th}\left(z_{n},z_{m},Th_{old},Th_{new}\right)+\alpha \;\left(R_{Th}+\gamma \left(max\;\left(Q_{Th}\left(z_{n},z_{m},Th_{old},Th_{new}\right)\right)\right)\right)$  **9.** $Threshold\left(z_{n},z_{m},Th_{old}\right)=Th_{new}$
 **10.** Increase the time step *n* by 1**3.** End while

### 3.2. Second Learning Hierarchy Level: Learning the Best Helper

As discussed in [29], the threshold reveals a fog fragment allowed boundary of using the neighbors’ helpers and feasible internal devices in emergencies (i.e., if a fragment does not have a feasible device with a priority lower than the threshold, the fog node can receive help from its neighbors). Therefore, in the second learning hierarchy level, the learner’s problem is selecting the best device as an emergency helper from its own feasible devices and neighbors’ feasible lowest priority devices in their help lists. In this part, we explain features that are added or changed in this study to complete the model from [29] as the second learning hierarchy level.

#### 3.2.1. State

A Boolean array, *flag*, presents situations of *n_d_* connected devices to the learner, numbered from 1 to *n_d_*, in the time step *n.* When one of its elements is 0, the connected device is in a normal situation, and when it is 1, an emergency occurs for that device. In every time step, the *flag* is updated based on the device emergency reports. As an emergency situation becomes normal, the related index of *flag* becomes 0. Due to implementation limitations in large state spaces, unlike [29], the learner state is not its *flag* array. In this study, the learner’s state space has four dimensions. The first dimension indicates the number of devices that recently encountered emergencies, and the device number is the same as its index in the *flag* array. The second and third dimensions are *z_n_* and *z_m_*, and the fourth dimension is the current threshold, which is the result of the last learning hierarchy level based on the second and third dimensions (i.e., the combination of part of the state and selected action of the first learning hierarchy level (*z_n_*, *z_m_*, and *Th_new_*) becomes a part of the state of the second learning hierarchy level).

For emergency simulations in each time step, the uniformly distributed pseudo-random integer function generates *n_d_* values randomly from {0, 1} for *n_d_* elements of the *flag* to demonstrate the device situations in the learner fragment in [29]. Emergency events do not occur uniformly for all fragment devices, as some devices are more exposed to emergency events and have high assigned priorities at the beginning of their smart environment life. Thus, in this study, we improved the method of generating 1 for *flag* elements (emergencies) so that encountering emergencies increases based on the IoT device priorities in a fragment. In the simulation, white priority devices never face emergencies, and with probabilities of 20%, 40%, 60%, and 80%, the *flag* elements of green, yellow, orange, and red priority devices become 1 to indicate emergencies.

#### 3.2.2. Action

As in [29], learning is performed in emergencies, and action is only selected if an IoT device has an emergency; therefore, the learner should perform an appropriate action to prepare the required bandwidth for the emergency device (*d_i_*). Moreover, the *i*^th^ element of the *flag* array was recently changed to 1 at the beginning of the current time step because the index of this emergency device is *i*. Therefore, the learner’s selected action is a device (*d_k_*) from its own feasible device set or the neighbors’ feasible devices in their help lists to assign the helper’s bandwidth to *d_i_* as needed. Action *a_i_* determines *k*, demonstrating that a device with index *k* is selected to help the emergency device with index *i*.

#### 3.2.3. Policy

The policy of the second learning hierarchy level is selecting helpers among its own or neighboring feasible devices so that the punishment value becomes the minimum, whereas the reward value becomes the maximum.

#### 3.2.4. Reward Function

As in [29], the Boolean array *track* includes *n_d_* elements, which become 1 when the related devices receive help during an emergency. In the following, *fnort* is formulated so that if the learner has helped an emergency device or both related elements of *flag* and *track* are 0, the related element of *fnort* becomes 1. When the learner cannot help a needy device, the related element of *fnort* remains 0. The only modification of the reward function compared to [29] is the part using *fnort*. The *R_nQ_* function providing the reward value of the second-level algorithm of the learning hierarchy is defined in Equation (6):(6)$$R_{nQ}=operation-punish\left(i\right)-penalty-blame,$$
where *operation* is calculated via Equation (7):(7)$$operation=operation+\sum_{i}^{n_{d}}\left(fnort\left(i\right)\times p\left(i\right)\right).$$

If no device could help in the learner fragment due to the learner’s past poor operation, the *penalty* decreases *R_nQ_* by the number of devices with lower priorities than the needy device’s priority in the learner fragment, not just the number of devices in the feasible device set. If the learner selects a helper for the current needy device (*d_i_*) among its own feasible devices, *punish*(*i*) is the number of devices in the feasible device set with priorities lower than the threshold and the selected helper’s priority. If the learner chooses a neighbor’s helper, the value of *blame* equals the number of owned feasible devices with lower priorities than the threshold. Algorithm 2 summarizes the second-level algorithm of the learning hierarchy in which successful bandwidth management (*SBM*) is the final comparison criterion [29].

**Algorithm 2:** Second-level algorithm of the learning hierarchy.**1.** Initialization**2.** While not converged, do **1.** For each element *i* of the learner’s *flag*, do  **1.** If *flag(i)* is converted from 1 to 0, do    Find device *u* that helped device *i* among all devices, including neighbors’ helpers, and return the borrowed bandwidth from *d_i_* to *d_u_*  **2.** Else if *flag(i)* is converted from 0 to 1, do   **a.** Make a feasible device set, including the nodes’ own feasible devices and neighbors’ feasible help lists.   **b.** With probability ɛ, randomly choose device *k* among the feasible device set, otherwise:      $a_{i}=argmax_{j\in \;feasible}\left(Q\left(i,z_{n},z_{m},Th_{new},j\right)\right)$, where *j* is the index of the selected.   **c.** Increase the bandwidth of *d_i_* and decrease the bandwidth of *d_j_*, as much as *d_i_* needs.   **d.** Calculate *penalty* or *punish(i)* or *blame(i)* based on the learner’s action and then calculate *R_nQ_*:       $R_{nQ}=operation-punish\left(i\right)-penalty-blame$
   **e.** Update Q matrix:       $Q_{new}\left(i,z_{n},z_{m},Th_{new},a_{i}\right)=\left(1-\alpha \right)Q_{old}\left(i,z_{n},z_{m},Th_{new},a_{i}\right)+$      $\alpha \;\left(R_{nQ}+\gamma \left(max_{k^{\prime }\epsilon \;feasible}\;\left(Q_{old}\left(i^{\prime },z_{n},z_{m},Th_{new},a_{i}^{’}\right)\right)\right)\right)$
  **3.** End if **2.** End for **3.** Make the next *flag* based on the learner’s device priorities. **4.** Repeat lines 3 to 26 for neighboring fog nodes **5.** Calculate *SBM_n_* using *countpun* considering all fragments **6.** Calculate the average *SBM* **7.** Increase the time step *n* by 1**3.** End while

### 3.3. Results

Figure 4 presents the process results of the first learning hierarchy level. Figure 5 displays a better view of Figure 4 to gain a better perception of the learning process. The vertical and horizontal axes denote the average of *R_Th_* and the time steps, respectively. Over time, the learning process progresses, and the learner experience is reinforced. Therefore, the learner’s selections improve, and the average reward increases. Finally, the learner converges to the highest possible average reward. The learning process becomes stable after convergence. When the learning process is complete, the learner selects the best threshold level based on its experiences according to the received rewards.

When the first learning hierarchy level is completed, the resulting matrix, *Threshold*, is used by the second learning hierarchy level. Figure 6 illustrates the results in comparison to the scenario in [29]. The results reveal that making the threshold adaptive via HRL results in better performance from the average reward and time perspectives, in contrast to [29], which solved the cooperation problems using a fixed threshold with RL. Table 3 describes the final results of both scenario simulations.

## 4. Conclusions

Due to the network complexity and uncertainty of emergency distribution parameters, we propose RL as a powerful machine learning approach without a trainer or supervisor to meet various needs in constantly expanding smart networks with limited bandwidth. In this study, we continued our previous work on fog fragment cooperation in bandwidth management under a predefined threshold constraint in a smart environment with limited shared bandwidth among several fog fragments. The IoT devices of these smart fragments may experience uncertain emergencies regarding the time and sequence needed for more bandwidth for further higher-level communication. We promote fog fragment cooperation by completing the cooperation qualification and using an HRL approach to the adaptive threshold. Although equipping the method to the adaptive threshold and restricting the fog fragment cooperation make the learning procedure more difficult, the HRL approach increases the network performance. Moreover, despite two levels of learning, the proposed approach has less total convergence time in the second hierarchy level.

The proposed approach applies to smart environments where fixed priorities and bandwidths are assigned to the devices. If these setups are changed, the learning must be restarted. Moreover, it is assumed that all fragments of this study are the same in terms of priority (i.e., no fragment has a higher priority than the others, whereas the importance of different fragments of a smart environment is not the same). It would be valuable to assign different priorities to fragments and add another learning level to consider the fragment priorities in future work.

## Figures and Tables

**Figure 1 sensors-21-07053-f001:**
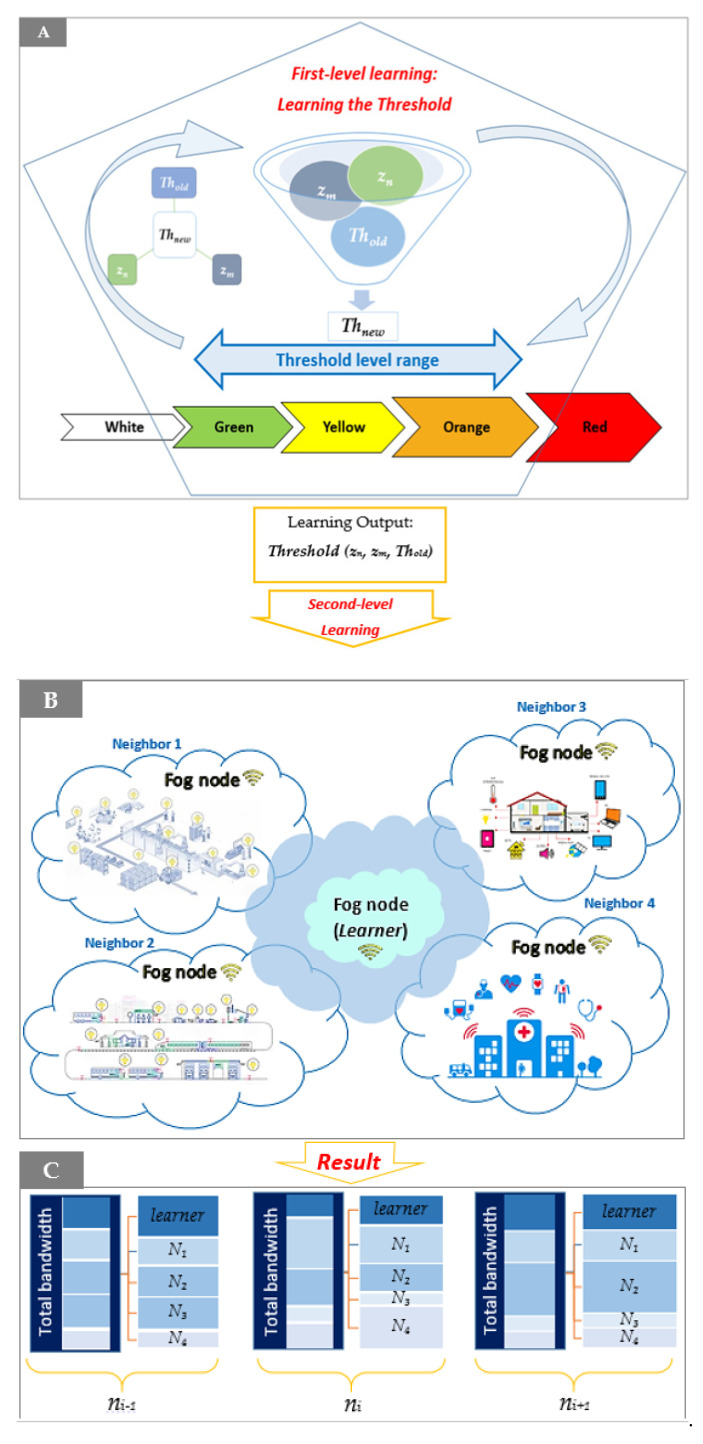
Illustration of the two-level learning hierarchy in which (**A**,**B**) denote the first (lower) and second (higher) learning hierarchy levels. The second learning level uses the final threshold matrix, the result of the first level after convergence. (Part **C**) presents the amount of each fragment’s bandwidth during three consecutive time steps.

**Figure 2 sensors-21-07053-f002:**
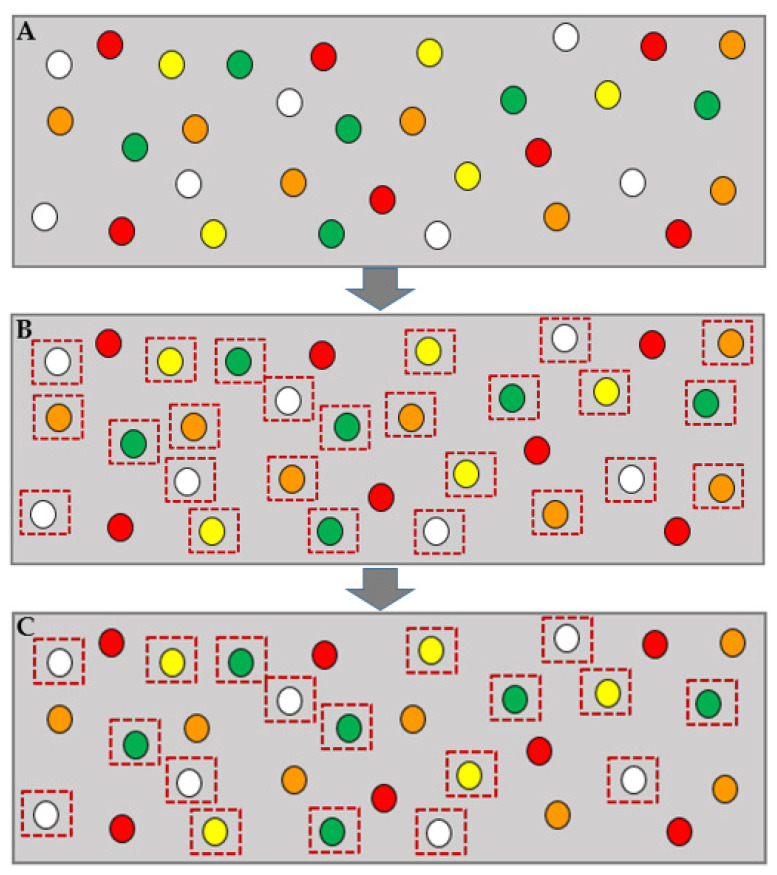
Feasible device set forming. (**A**) illustrates a learner fragment. Considering all IoT devices have enough bandwidth to help, feasible devices are specified in (**B**). Assuming the current emergency device’s priority is orange and *Th =* yellow (2), the feasible device set includes devices with priorities lower than yellow priority (**C**).

**Figure 3 sensors-21-07053-f003:**
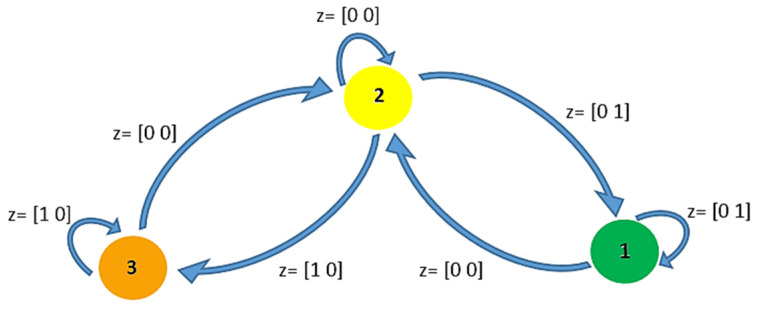
Transition states among green, yellow, and orange as threshold levels, perceiving $z=[z_{n}\;z_{m}]$ as the internal and external situations.

**Figure 4 sensors-21-07053-f004:**
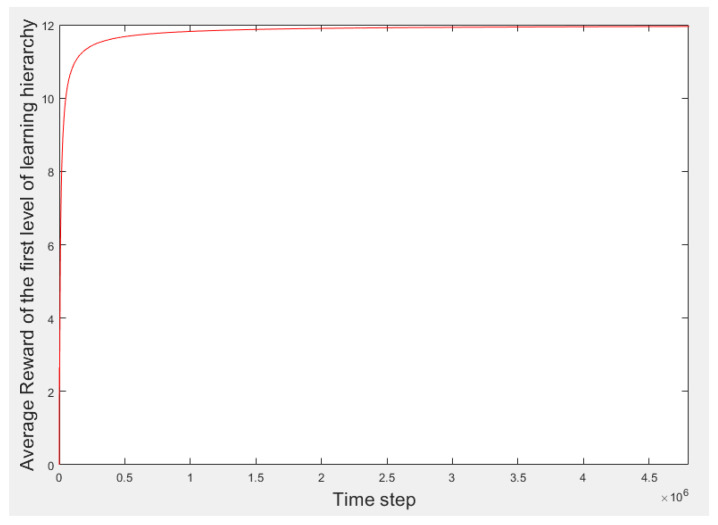
Results of the first learning hierarchy level.

**Figure 5 sensors-21-07053-f005:**
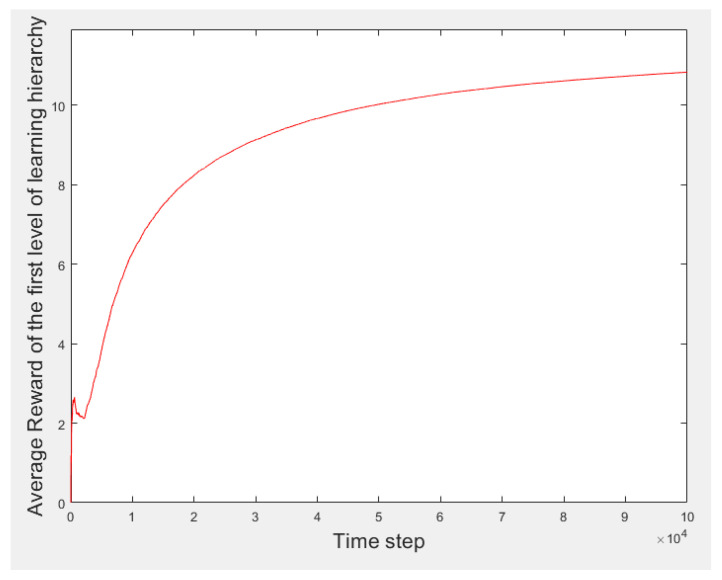
Initial part of first-level learning results before the process converges.

**Figure 6 sensors-21-07053-f006:**
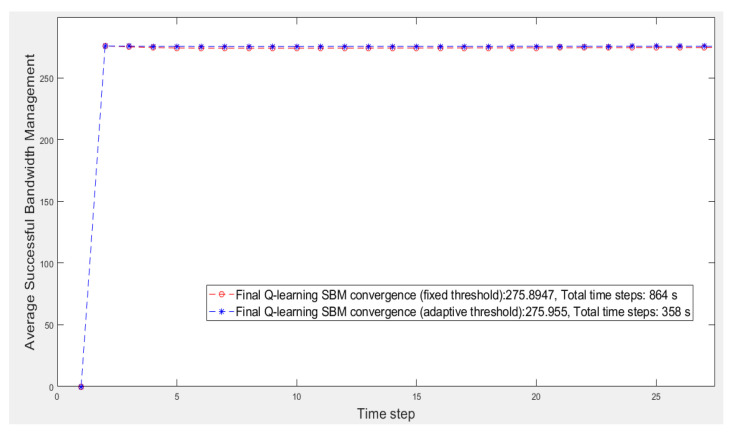
Second learning hierarchy level results compared to learning with a fixed threshold.

**Table 1 sensors-21-07053-t001:** Learning policy.

	*z*		Policy	
	*z_n_*	*z_m_*	Decreasing *Th_old_*	Helping Neighbors
1	0	0	0	0
2	0	1	0	1
3	1	0	1	0
4	1	1	1	1

Note: 0 and 1 represent yes and no, respectively, in the policy columns.

**Table 2 sensors-21-07053-t002:** Policy goals.

	State			Action
	*z_n_*	*z_m_*	*Th_old_*	*Th_new_*
1	0	0	X	2
2	0	1	1,2	1
			3	2
3	1	0	1	2
			2,3	3
4	1	1	1	1
			2,3	3

Notes: Threshold levels 1, 2, and 3 represent green, yellow, and orange, respectively; *X* denotes levels 1, 2, and 3.

**Table 3 sensors-21-07053-t003:** Simulation results.

	Fixed Threshold Learning with Neighbor Cooperation	Adaptive Threshold Learning with Neighbor Cooperation
**Final *SBM* convergence:**	275.89	275.95
**Total time steps:**	864 s	358 s

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
