# Peer review of "Toward an Adaptive Threshold on Cooperative Bandwidth Management Based on Hierarchical Reinforcement Learning"

_sensors, 2021, doi:10.3390/s21217053_

Round 1

Reviewer 1 Report

Authors have worked on bandwidth management in a smart environment with several fog fragments using limited shared bandwidth, where IoT devices may experience uncertain emergencies in term of the time and sequence need for more bandwidth for further higher-level communication. The authors claim to have introduced fog fragment cooperation using an RL approach under a pre-defined threshold constraint. 

  • Abstract should be revised to summarize the whole work in a ideal way.
  • Introduction should be revised: (1) Introduce the problem (2)discuss about some of the existing solutions (3)identify the gap or scope of improvement (4) discuss in order to address the identified gaps what is the methodology used (5) list out the contributions (6) Organization of paper.
  • "Materials and Methods" section should be written in such a flow which should be clearly understandable by readers.
  • "Related work" Section is missing which should include review of state-of-the-art approaches. Authors can consider referring the following article:
    • Fault-Resilience for Bandwidth Management in Industrial Software-Defined Networks
  • Mathematical modelling should be carried out for the proposed approach.
  • The approach should be compared with an existing approach and detailed result analysis should be provided.
  • Limitations of the proposed approach should be written in "Conclusion".
  • Proof-reading is required.

Reviewer 2 Report

Dear Authors,

The content of the paper fits perfectly into the scope of a Sensors journal.

The 40 publications analyzed in the Introduction (the initial sections of the article were the basis for achieving the above-mentioned goal.

Towards the end of the paper, there are statements about the important contribution of its results for on cooperative bandwidth management based on hierarchical reinforcement learning.

The article contains some new data.

The text is clear and easy to read.

Some specific comments on the paper:

  • To what extent do fog fragments influence the final simulation results? Are there other parameters of influence, are they important and anticipated in future studies, and how are they accounted for in the models presented?
  • Are the parameters in Table 5 optimal? To what extent has this been justified?

Reviewer 3 Report

This document is well written and presented. There are some minor comments:

1) There are some typographical errors. See, for example, lines 38, 320-321,
etc.
2) The equations required variable declarations just before or after them. For example,
the parameter ND is not defined in (1), the definition of p(i) is missing too, etc.
Line 454, n_d can be confused with ND, etc. Probably, an acronym table will
be welcome.

3) Table 2 is fragmented in two pages.

4)
The algorithm of Table 3 in its line 12-13, the expression R_th(n)
does not agree with equation (4). Therefore, the algorithms require
some minor revisions.

5)
The situation in row 4 of Table 1 is said to be not allowed and not logical
too. For initial readers on the topic, the same could be wrongly concluded
for the first row of Table 1.

Round 2

Reviewer 1 Report

Proof-reading is required.

Author Response

Dear reviewer Proof-reading is done. Please see the attachment. 
